# Isotropic Representation Can Improve Zero-Shot Cross-Lingual Transfer on Multilingual Language Models

**Yixin Ji[1], Jikai Wang[1], Juntao Li[1*], Hai Ye[2], Min Zhang[1]**
[1]Institute of Computer Science and Technology, Soochow University, China
[2]Department of Computer Science, National University of Singapore
{jiyixin169, risus254}@gmail.com;
yehai@comp.nus.edu.sg;
{ljt,minzhang}@suda.edu.cn

## Abstract

With the development of multilingual pre-trained language models (mPLMs), zero-shot cross-lingual transfer shows great potential. To further improve the performance of cross-lingual transfer, many studies have explored representation misalignment caused by morphological differences but neglected the misalignment caused by the anisotropic distribution of contextual representations. In this work, we propose enhanced isotropy and constrained code-switching for zero-shot cross-lingual transfer to alleviate the problem of misalignment caused by the anisotropic representations and maintain syntactic structural knowledge. Extensive experiments on three zero-shot cross-lingual transfer tasks demonstrate that our method gains significant improvements over strong mPLM backbones and further improves the state-of-the-art methods.[1]

## 1 Introduction

Cross-lingual transfer aims to utilize the rich semantics and syntactic knowledge in high-resource source languages to improve the performance of low-resource target languages. Benefiting from the "scaling effect", language models pre-trained on hundreds of languages have dominated cross-lingual transfer for years owing to their superior performance and generalization capability, which can learn a unified representation space through self-supervised learning (Devlin et al., 2019; Conneau and Lample, 2019; Conneau et al., 2020a; Xue et al., 2020; Chi et al., 2021, 2022; Scao et al., 2022a). These powerful multilingual pre-trained language models (mPLMs) not merely improve task performance with full supervised learning but even the zero-shot cross-lingual transfer (Wu and Dredze, 2019a; Hsu et al., 2019; Li et al., 2021; Sherborne and Lapata, 2022).

---

*Corresponding author.

[1]Our code is available at https://github.com/Dereck0602/IsoZCL.

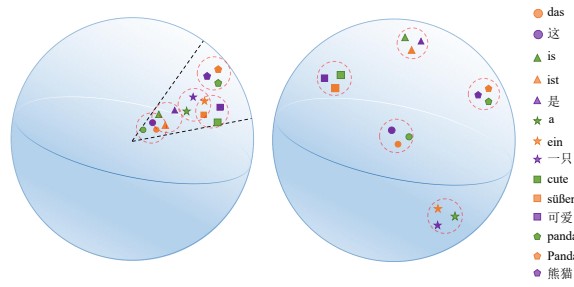

(a) anisotropic representations  (b) isotropic representations

Figure 1: An illustration of anisotropic and ideally isotropic multilingual representations.

The core of cross-lingual transfer is to align representations among different languages (Lample et al., 2018b; Cao et al., 2020; Pan et al., 2021; Dou and Neubig, 2021). Existing mPLMs can align the representation well for myriads of the cross-lingual transfer scenarios but fail to handle these language pairs differed significantly under the zero-shot setting, especially for languages with distinct morphological features (Ahmad et al., 2019a,b). Therefore, many works have explored the key factors affecting the alignment of language representations (Pires et al., 2019; Karthikeyan et al., 2020; Libovický et al., 2019; de Vries et al., 2022) and proposed solutions accordingly (Cao et al., 2020; Chi et al., 2021; Pan et al., 2021; Zhao et al., 2021; Huang et al., 2021b). Existing methods can be roughly divided into three categories: 1) using parallel corpus (Chi et al., 2021; Wei et al., 2021; Feng et al., 2022) or bilingual dictionary (Cao et al., 2020; Qin et al., 2021) to better align contextualized word embedding spaces; 2) utilizing morphological or syntactic features (Ahmad et al., 2021; Yu et al., 2021; Zhao et al., 2021) to eliminate misalignment; 3) leveraging robust training methods (Huang et al., 2021b) to tolerate misaligned representations.

However, additional parallel corpora are difficult to obtain for many extremely low-resource

languages, while annotating morphological or syntactic features requires considerable human effort. Though the robust training methods are free from additional supervision signals, they ignore the misalignment caused by the distributional properties of representation. As found by Rajaee and Pilehvar (2022), the representation distribution of mPLMs is highly anisotropic, where most words are squeezed into a narrow region in the representation space. As shown in Figure 1a, even semantically irrelevant representations can appear in the neighborhood. Thus, there are reasons to believe that highly anisotropic representations hurt cross-lingual alignment in the representation space.

To alleviate the aforementioned obstacles, we propose an isotropy enhancement strategy that can improve the representation alignment at the semantic level while maintaining the essential syntactic knowledge, which is indispensable for cross-lingual transfer (Ahmad et al., 2021). As an excessively isotropic representation space is risky to pull apart representations that should be aligned, we also introduce a constrained code-switching method to better utilize the readily available bilingual dictionaries. To verify the effectiveness of our proposed method, we launch experiments on three zero-shot cross-lingual tasks, i.e., paraphrase identification, natural language inference, and sentiment classification. Experimental results demonstrate that our proposed method significantly improves the performance of zero-shot cross-lingual transfer upon strong mPLMs. Further analytical exploration confirms that our method can alleviate the anisotropy problem of pre-trained representations while preserving syntactic knowledge implicit in the representations as much as possible.

## 2 Preliminary

In this section, we briefly introduce the anisotropic problem of representation learning and the risk of undermining knowledge of syntactic structures in existing methods of mitigating anisotropy.

### 2.1 Anisotropic Problem of Contextual Representations

Anisotropy is a geometrical property of contextual representations. As defined by Li et al. (2020), anisotropic representations occupy a narrow cone in the vector space. Conversely, isotropic representations are uniformly dispersed in the vector space. It is widely believed that anisotropy lim-

its the expressiveness of contextual representations (Gao et al., 2019; Wang et al., 2020; Li et al., 2020; Su et al., 2021; Rajaee and Pilehvar, 2021a). Next, we will introduce two different metrics for measuring isotropy quantitatively.

**Cosine Similarity.** Since the word vectors are squeezed together, the external manifestation of anisotropic representations is that for any two words, the cosine similarity is large. If representations are isotropic, cosine similarities of random representations are close to zero (Gao et al., 2019; Ethayarajh, 2019). The metric can be formulated as follows:

$$I_{Cos}(\mathcal{W}) = \frac{1}{N} \sum_{i,j,x_i \neq x_j}^{N} \text{Cos}(x_i, x_j) \qquad (1)$$

where $x_i$, $x_j$ are randomly sampled representations. $N$ is the number of sampled representation pairs. $I_{Cos}(\mathcal{W})$ closer to 0 indicates that the representations are more isotropic.

**Principal Components.** Following Mu and Viswanath (2019), we use a partition function (Arora et al., 2016) to measure the isotropy:

$$F(u) = \sum_{i=1}^{N} \exp\left(u^T w_i\right) \qquad (2)$$

where $w_i \in W$ is a contextual word embedding, $N$ is the number of embeddings in the representation space, $u \in U$ is the eigenvector of the embedding matrix $W^T W$. According to Arora et al. (2016), if representations are isotropic, $F(u)$ could be approximated using a constant. Thus, Mu and Viswanath (2019) propose a metric based on principal components:

$$I_{PC}(\mathcal{W}) \approx \frac{\min_{u \in U} F(u)}{\max_{u \in U} F(u)} \qquad (3)$$

$I_{PC}(\mathcal{W})$ closer to 1 indicates representations are more isotropic.

### 2.2 Syntactic Knowledge Probing of Existing Isotropy Enhancement Methods

Currently, research on enhancing the isotropy of contextual representations focuses on feature-based learning or training from scratch. For fine-tuning the pre-trained language model, although the fine-tuned model still has severe anisotropy, directly applying existing methods to enhance isotropy cannot effectively improve the performance and may even degrade it (Rajaee and Pilehvar, 2021b; Zhang

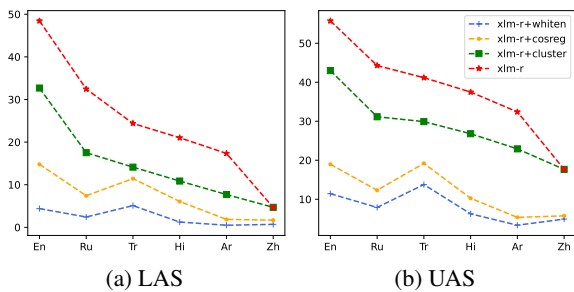

(a) LAS        (b) UAS

Figure 2: Depprobe results on the vanilla fine-tuned model and three isotropic enhancement methods.

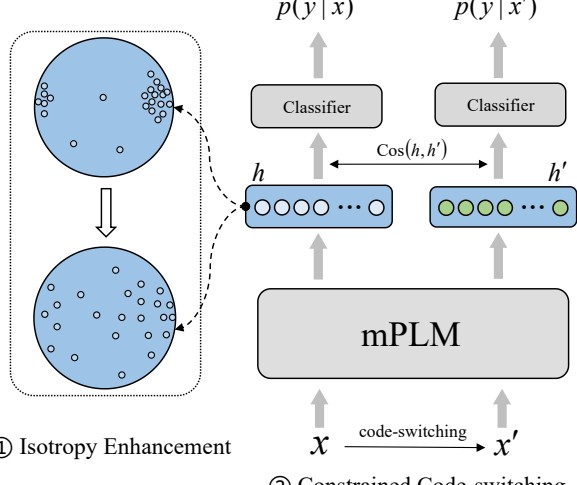

① Isotropy Enhancement

② Constrained Code-switching

Figure 3: Overview of our proposed method which includes isotropy enhanced fine-tune and constrained code-switching.

et al., 2022). Because the fine-tuning has changed the distribution of linguistic and task-specific representations in the pre-trained language model, existing methods may destroy task-essential knowledge (Rajaee and Pilehvar, 2021b).

We analyze that most methods are applied to semantic-level tasks, such as sentence representation and semantic textual similarity. Thus they neglect to consider the structural syntactic knowledge implicit in the representation. However, structured syntactic knowledge implicit in representations is essential for many natural language understanding tasks, especially for cross-lingual transfer (Ahmad et al., 2021). To verify our suspicions, we use the recently proposed Depprobe (Müller-Eberstein et al., 2022), a linear probe that can extract labeled and directed dependency parse trees from contextual representations, to measure the structural knowledge of fine-tuned representations.

In experiments, we train the Depprobe on the English treebank and evaluate on six target languages. We report two representative metrics in dependency parsing, labeled attachment scores (LAS) and unlabeled attachment scores (UAS), which measure the accuracy of predicted dependency graphs. All these datasets are from Universal Dependencies v2.8[2]. The mPLM used has been fine-tuned on the XNLI dataset. We compare the impact of three representative isotropic enhancement methods (whitening transformation (Su et al., 2021), cluster-based methods (Rajaee and Pilehvar, 2021a), and CosReg (Gao et al., 2019)) on structural knowledge. Figure 2 shows the LAS and UAS of six languages on XLM-R. We can observe that all these methods lead to a sharp drop in the results of the syntactic probe experiments. Therefore, it is necessary to devise an isotropic enhanced method for pre-trained models while preserving knowledge at semantic

and structural levels.

## 3 Method

In this section, we introduce our method for zero-shot cross-lingual transfer, which comprises of isotropy enhancement and constrained code-switching. Figure 3 shows the overview of our proposed method. Before elaborating our method in detail, we first present some necessary notations. Given the training dataset $\mathcal{D} = \{(x_i, y_i)\}_{i=1}^n$, the loss of fine-tuning can be written by:

$$\mathcal{L}_{\mathcal{D}}^{ft} = \sum_{(x_i, y_i) \in \mathcal{D}} l(f(x_i; \theta), y_i) \qquad (4)$$

where $x_i$ is a token sequence (e.g., a sentence), and $y_i$ is its label. $f(\cdot; \theta)$ is the model to be optimized, and $l(\cdot; \cdot)$ is the loss function to learn a task-specific model on the source language. For classification tasks, $l(\cdot; \cdot)$ is usually the cross-entropy loss. We denote the contextual representation of $x_i$ as $\boldsymbol{h}_i \in \mathbb{R}^{L \times d}$, where $L$ is the length of the sample and $d$ is the dimension of the mPLM.

### 3.1 Isotropy Enhancement

To enhance the isotropy of the token representation space of the multilingual pre-trained language model and meanwhile maintain semantic and syntactic features as much as possible, we introduce an isotropy-aware loss as a regular term to force the representation distribution $\boldsymbol{H}$ to be close to an isotropic distribution. For feasibility, we suppose $\boldsymbol{H}$ obeys a normal distribution $\mathcal{N}(\boldsymbol{\mu}, \boldsymbol{\Sigma})$, which is

---
[2]http://hdl.handle.net/11234/1-3687

a reasonable assumption and has also acquiesced in other methods such as whitening transformation. As we all know, the zero-mean isotropic normal distribution $\mathcal{N}(\mathbf{0}, \sigma^2 \mathbf{I})$ is a perfect isotropic distribution, thus we can use the Wasserstein distance between the two normal distributions to measure the isotropy degree of $\mathbf{H}$ (Salmona et al., 2021):

$$\mathcal{W} = \|\boldsymbol{\mu}\|_2^2 + Tr(\boldsymbol{\Sigma} + \sigma^2 \mathbf{I} - 2\sigma^2 \boldsymbol{\Sigma}^{\frac{1}{2}}) \quad (5)$$

When $\mathcal{W}$ is used as a regular term, to ensure the stability of the optimization process, the value of $\mathcal{W}$ should be on the same order of magnitude as $\mathcal{L}_{\mathcal{D}}^{ft}$ by setting the $\sigma$ to a small value. Therefore, $\sigma$ is a very sensitive hyperparameter. To avoid tedious hyperparameter search, we calculate the Wasserstein distance between the normalized representation distribution and $\mathcal{N}\left(\mathbf{0}, \frac{1}{d}\mathbf{I}\right)$, which has been proved to be equivalent to (5) by Fang et al. (2023):

$$\mathcal{W} = \|\boldsymbol{\mu}\|_2^2 + 1 + Tr(\boldsymbol{\Sigma}) - \frac{2}{\sqrt{d}}Tr(\boldsymbol{\Sigma}^{\frac{1}{2}}) \quad (6)$$

For a given batch of samples, the mean and covariance matrix of representations are as follows:

$$\boldsymbol{\mu} = \frac{1}{N}\sum_{i=1}^{N}\frac{\boldsymbol{h}_i}{\|\boldsymbol{h}_i\|_2} \quad (7)$$

$$\boldsymbol{\Sigma} = \frac{1}{N}\sum_{i=1}^{N}\left(\frac{\boldsymbol{h}_i}{\|\boldsymbol{h}_i\|_2} - \boldsymbol{\mu}\right)^T\left(\frac{\boldsymbol{h}_i}{\|\boldsymbol{h}_i\|_2} - \boldsymbol{\mu}\right) \quad (8)$$

where $\|\cdot\|_2$ denotes $L_2$ norm, $N$ is the number of token representations in the given batch. The proposed isotropy-aware loss can be formulated as:

$$\mathcal{L}^{iso} = \lambda_1 \mathcal{W} \quad (9)$$

where $\lambda_1$ is a hyperparameter which controls the degree of isotropy.

## 3.2 Constrained Code-switching

Isotropy enhancement provides a suitable representation space property for multilingual alignment, making unrelated representations not misaligned due to representation space degradation. However, an excessively isotropic representation space is risky to pull apart representations that should be aligned. Thus, we align representations with similar semantics by constraining code-switch data. Code-switching means that more than one language alternates in a sentence, which is widely considered to provide anchor points for aligning multilingual

representations (Conneau et al., 2020b; Yang et al., 2020; Qin et al., 2021). We randomly select words in the source language text, query bilingual dictionaries, and replace them with target language words to obtain code-switching data. More details can be found in §4.2.

**Consistency Constraints.** We denote the original and corresponding code-switching data as $x = \{w_1, w_2, \ldots, w_L\}$ and $x' = \{w_1, w_2', \ldots, w_L\}$, where $w_i'$ means the replaced source language token by target languages. For a sample $x_i \in \mathcal{D}$ and its code-switching data $x_i' \in \mathcal{D}'$, the mPLM will produce two different representations $\boldsymbol{h}_i, \boldsymbol{h}_i' \in \mathbb{R}^{L \times d}$. To further promote the alignment among representations with similar semantic in different languages, we utilize the following consistency constraints between the original and its code-switching data:

$$\mathcal{L}^{reg} = \lambda_2 \sum_{\substack{x_i \in \mathcal{D} \\ x_i' \in \mathcal{D}'}} \text{Cos}(\boldsymbol{h_i}, \boldsymbol{h_i}') \quad (10)$$

where $\text{Cos}(\cdot, \cdot)$ is the cosine similarity between two representations and $\lambda_2$ is a hyper-parameter.

## 3.3 Training

As mentioned above, the whole training process is divided into two stages. Our method first improves the spatial distribution of multilingual representations via isotropic enhancement, and then promotes representation alignment via constrained code-switching. For the first stage, the loss function takes the form of:

$$\mathcal{L}_1 = \mathcal{L}_{\mathcal{D}}^{ft} + \mathcal{L}^{iso} \quad (11)$$

where $\mathcal{L}_{\mathcal{D}}^{ft}$ is to learn a task-specific model on the source language. $\mathcal{L}_{iso}$ is to enhance the isotropy of contextual representations during fine-tuning by narrowing the difference between the representation space and the standard normal distribution. In the second stage, the loss function is:

$$\mathcal{L}_2 = \frac{1}{2}(\mathcal{L}_{\mathcal{D}}^{ft} + \mathcal{L}_{\mathcal{D}'}^{ft}) + \mathcal{L}^{reg} \quad (12)$$

where $\mathcal{L}_{\mathcal{D}}^{ft}, \mathcal{L}_{\mathcal{D}'}^{ft}$ are the fine-tuning loss on original data and code-switching, respectively, $\mathcal{L}^{reg}$ is for aligning constraints between the representations of code-switching samples and the original data. We split training into two stages for two reasons. First, these losses may interfere with each other because isotropy enhancement widens the

distance between representations so that representations are distributed more evenly, while constrained code-switching makes multilingual representations with the same semantics close. When optimizing, achieving a delicate balance is challenging. Also, simultaneous optimizations add more cost (Eq.5 has more computational cost than Eq.10).

## 4 Experiments

### 4.1 Datasets

We conduct our experiments on three different cross-lingual tasks, including paraphrase identification (PAWS-X; Yang et al., 2019), natural language inference (XNLI; Conneau et al., 2018) and sentiment classification (MARC[3]; Keung et al., 2020). For MARC, following Keung et al.(2020), we splice the "review title" and "product category" after the review. Due to the limitations of computing resources, we sampled 25% of the original training set as training set. The data characteristics are shown in Appendix A.

### 4.2 Experimental Setup

Our experiments are based on two mPLMs, mBERT (Devlin et al., 2019) and XLM-R-large (Conneau et al., 2020a). On each task, we fine-tune the mPLM for five epochs with batch size 32 on the English training set, select the best model on the English development set, and then evaluate the cross-lingual performance on test sets of all target languages. We run 2 epochs in the first stage and 3 epochs in the second stage for PAWS-X and MARC. For XNLI, the epochs of the two stages is 1 and 4. For PAWS-X and XNLI, we tune the learning rate in {1e-6, 2e-6, 5e-6, 1e-5, 2e-5}; for MARC, learning rate in {8e-7, 1e-6, 2e-6, 1e-5, 2e-5}. We tune coefficient $\lambda_1$ and $\lambda_2$ in {0.5, 1.0}. We report details of all hyper-parameters in Appendix B. When constructing code-switching data, we set the probability that each token in a sample is replaced with a target language token to 0.5. The bilingual dictionaries we used are from MUSE (Lample et al., 2018a)[4]. We report the average score on the test set of 5 runs with different seeds. We conduct the experiments on one NVIDIA GTX3090 GPU.

---

[3]https://github.com/awslabs/open-data-docs/blob/main/docs/amazon-reviews-ml/license.txt
[4]https://github.com/facebookresearch/MUSE

### 4.3 Baselines

We compare our methods with the following isotropy enhancement methods and strong zero-shot cross-lingual transfer baselines:

**Fine-tune.** We fine-tune all the parameters of the mPLM with the English training set and then evaluate on test sets of all target languages.

**BN.** Zhao et al. (2021) propose a vector space normalization method. They apply batch normalization to the last layer representations of mPLM to induce language-agnostic representations and increase the discriminativeness of embeddings.

**IsoBN.** Zhou et al. (2021) explore the isotropy of the pre-trained [CLS] embeddings and propose isotropic batch normalization (IsoBN). They assume that the absolute correlation matrix of embeddings is block-diagonal.

**CosReg.** The high cosine similarity between word representations is an extrinsic indication of representation degeneracy. Thus, Gao et al. (2019) add a CosReg loss to minimize the cosine similarities between any two contextual token embeddings.

**NoisyTune.** Wu et al. (2022) inject noise into the parameters of the pre-trained model when fine-tuning, preventing the pre-trained model from over-fitting on the source language.

**DA.** Huang et al. (2021b) improve zero-shot cross-lingual transfer through robust training based on data augmentation. They use a predefined synonym set to generate augmentation examples. For a fair comparison with our method, we set the number of augmented examples to 2 for all datasets.

### 4.4 Main Results

Table 1 shows zero-shot cross-lingual results on three datasets using two mPLMs. Following He et al. (2021), to better compare cross-lingual transfer to languages of different language families, we show the average results for three cases, including all languages (All), target languages other than English (Target), and non-Indo-European languages (Distant). We find that existing methods for enhancing isotropy in the fine-tuning stage do not significantly improve cross-lingual transfer performance. BN, IsoBN, and CosReg even perform worse than the vanilla fine-tuning on some datasets. In comparison, our method has consistent and significant performance gains on all these datasets and mPLMs. Especially for non-Indo-European languages, our method has better transfer ability. Then, our method achieves comparable or even better per-

| Models | PAWS-X | | | XNLI | | | MARC | | | Avg. |
|---|---|---|---|---|---|---|---|---|---|---|
| | **All** | **Target** | **Distant** | **All** | **Target** | **Distant** | **All** | **Target** | **Distant** | |
| mBERT | 83.51 | 81.77 | 76.22 | 66.34 | 65.20 | 61.43 | 45.80 | 42.68 | 38.18 | 65.22 |
| mBERT+BN | 82.99 | 81.14 | 75.40 | 66.39 | 65.23 | 61.40 | 44.43 | 40.98 | 36.39 | 64.60 |
| mBERT+IsoBN | 83.34 | 81.57 | 75.96 | 66.60 | 65.47 | 61.88 | 45.38 | 42.13 | 37.56 | 65.11 |
| mBERT+CosReg | 83.18 | 81.38 | 75.42 | 66.28 | 65.11 | 61.55 | 46.72 | 43.51 | 38.63 | 65.39 |
| mBERT+NoisyTune | 83.86 | 82.22 | 76.83 | 66.44 | 65.27 | 61.54 | 46.34 | 43.32 | 38.86 | 65.55 |
| mBERT+DA | 84.86 | 83.40 | 78.79 | 66.32 | 65.23 | 61.52 | 46.98 | 44.04 | 38.93 | 66.05 |
| mBERT+ours | 85.29 | 83.81 | 78.76 | 67.27 | 66.17 | 62.96 | 47.90 | 44.83 | 39.52 | 66.79 |
| mBERT+DA+ours | **85.50** | **84.19** | **79.67** | **67.47** | **66.51** | **63.19** | **48.21** | **45.34** | **39.73** | **67.06** |
| XLM-R | 87.01 | 85.63 | 81.09 | 79.35 | 78.71 | 76.94 | 58.88 | 57.52 | 53.99 | 75.08 |
| XLM-R+BN | 87.45 | 86.11 | 81.49 | 79.46 | 78.81 | 77.10 | 59.29 | 57.97 | 54.57 | 75.40 |
| XLM-R+IsoBN | 87.60 | 86.30 | 81.84 | 79.66 | 79.03 | 77.36 | 59.21 | 57.90 | 54.59 | 75.49 |
| XLM-R+CosReg | 87.73 | 86.42 | 81.66 | 79.27 | 78.63 | 76.90 | 59.46 | 58.11 | 54.55 | 75.49 |
| XLM-R+NoisyTune | 87.53 | 86.22 | 81.56 | 79.73 | 79.10 | 77.43 | 58.99 | 57.71 | 54.31 | 75.42 |
| XLM-R+DA | 88.86 | 87.74 | 83.89 | 81.10 | 80.60 | 79.10 | 59.48 | 58.05 | 54.28 | 76.48 |
| XLM-R+ours | 89.03 | 87.93 | 83.86 | 80.90 | 80.31 | 78.71 | 59.71 | 58.34 | 54.54 | 76.50 |
| XLM-R+DA+ours | **89.48** | **88.37** | **84.51** | **81.40** | **80.90** | **79.21** | **59.89** | **58.56** | **54.92** | **76.92** |

Table 1: Overall comparison of zero-shot cross-lingual performance between our proposed model and baseline models. **All** is the average result of all languages. **Target** is the average result of target languages other than English. **Distant** is the average result of non-Indo-European languages. **Avg.** is the average result of three datasets.

| Models | En | De | Es | Fr | Ja | Ko | Zh | Avg. |
|---|---|---|---|---|---|---|---|---|
| XLM-R | 95.57 | 89.77 | 90.19 | 90.57 | 80.30 | 79.80 | 83.17 | 87.01 |
| XLM-R+ours | 95.64 | **91.37** | **91.98** | **92.61** | **83.19** | **83.15** | 85.25 | **89.03** |
| only isotropy | **95.86** | 91.25 | 91.64 | 92.23 | 82.42 | 81.52 | 84.31 | 88.46 |
| only constrained | 95.43 | 90.90 | 91.05 | 91.25 | 82.38 | 82.50 | 84.13 | 88.23 |
| one stage | 95.78 | 91.00 | 91.46 | 92.29 | 82.06 | 81.50 | **85.41** | 88.50 |

Table 2: Ablation results on PAWS-X based on XLM-R. **only isotropy** means only using isotropy enhancement, **only constrained** means only using constrained code-swtching, and **one stage** means using isotropy enhancement and constrained code-switching together.

formance than the state-of-art model. Combining our method with DA can lead to a new state-of-art model. In conclusion, all the results confirm the effectiveness of our method for zero-shot cross-language transfer tasks. The results for each target language are detailed in the Appendix D.

## 5 Analysis and Discussion

### 5.1 Ablation Study

We conduct ablation experiments to analyze the contributions of isotropy enhancement, constrained code-switching and two-stage training. From the results in Table 2, we can see that both isotropy enhancement and constrained code-switching significantly impact the cross-lingual transfer performance. Using only isotropy enhancement consistently increases the performance of all languages in the test set, suggesting that anisotropic representations hurt both high-resource and low-resource languages. However, using only constrained code-switching sacrifices a little source language performance. We think this is because code-switching inevitably introduces some noisy samples. After removing the constrained code-switching, the performance of Japanese, Korean and Chinese, which belongs to a different language family than the source language, is severely degraded. It indicates that for target languages that are different from the source language, constrained code-switching can play a great role. We also investigate the necessity of two-stage training, and the experimental results provide a positive affirmation. We can observe that although the one-stage training has a significant performance improvement compared to the baseline, it still lags behind the two-stage training.

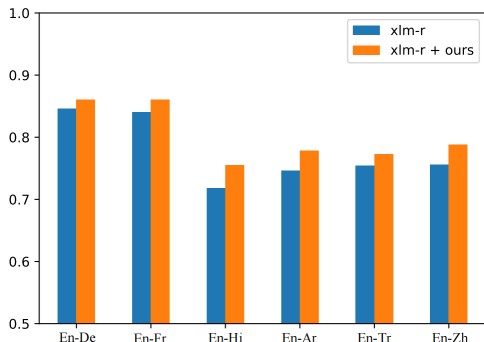

Figure 4: CKA scores on the XNLI test set.

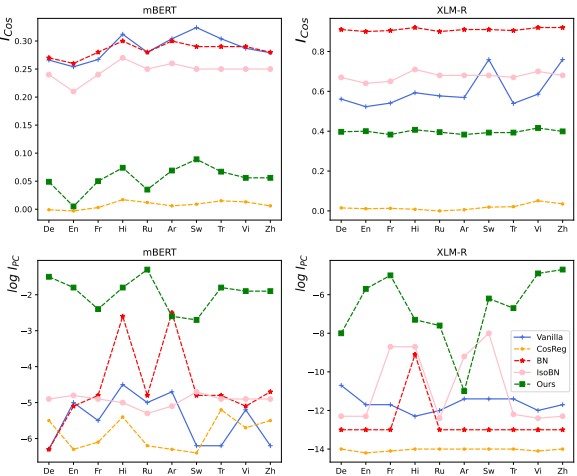

Figure 5: The isotropy metric on XNLI.

## 5.2 Cross-lingual Representation Discrepancy

Cross-lingual representation discrepancy quantitatively measures the degree of divergence between source and target language representations in the same embedding space. Yang et al. (2022) demonstrate that the cross-lingual transfer performance is highly related to the cross-lingual representation discrepancy. The smaller cross-lingual representation discrepancy correlates to better cross-lingual transfer performance. Following Conneau et al. (2020b), we utilize the linear centered kernel alignment (CKA) (Kornblith et al., 2019) score to indicate the cross-lingual representation discrepancy:

$$\mathrm{CKA}(X, Y) = \frac{\left\| Y^\top X \right\|_{\mathrm{F}}^2}{\left\| X^\top X \right\|_{\mathrm{F}}^2 \left\| Y^\top Y \right\|_{\mathrm{F}}^2} \qquad (13)$$

where X and Y are features of parallel sequences from the source and target languages. A higher CKA score means a smaller cross-lingual representation discrepancy. We use the cross-lingual representation discrepancy as a quantitative measure of sentence-level representation alignment. We conduct vanilla fine-tuning and our method on XNLI and evaluate CKA scores on test sets. Figure 4 shows CKA scores for six language pairs. We can observe that our method achieves a higher CKA score. Thus, we claim that our method helps induce better aligned multilingual representations.

## 5.3 Isotropy Measuring

To verify whether our proposed isotropy enhancement method can induce an isotropic representation space, we measure the isotropy of contextual representations on the test sets of XNLI using two metrics introduced in §2.1. As shown in Figure 5, whether it is $I_{Cos}$ or $I_{PC}$, our proposed method can obtain more isotropic contextual representations on mBERT and XLM-R model than the vanilla fine-

tuning and baselines. We note that although Cos-Reg can obtain a near-perfect $I_{Cos}$ by constraining the cosine similarity between word representations, it exhibits a high degree of anisotropy under the $I_{PC}$. Rajaee and Pilehvar (2021b) find cosine similarity might fail in high-dimensional space. Although the cosine similarity is close to zero, it's only isotropic in some dimensions and remains highly anisotropic from a global perspective. Therefore, we consider that using the CosReg method during fine-tuning does not inherently correct the anisotropic distribution of the representation.

## 5.4 Visualization

To demonstrate the effectiveness of our method on the representation distribution, we utilize PCA to reduce the contextual representations of XLM-R on PAWS-X to two dimensions[5]. Specifically, we randomly sample 10,000 token representations from the source language. As presented in Figure 6, each data point in the plots represents a contextual representation. We can observe from the left plot that after vanilla fine-tuning, most representations are concentrated in two regions of space, which reflects the highly anisotropic distribution of these representations. In contrast, the representations on the right plot are more evenly distributed throughout the space. The PCA visualization demonstrates that our isotropy enhancement method can produce a more uniformly distributed representation.

## 5.5 Syntax Probing

Through the discussion in §5.2, our method narrows the representation discrepancy between multi-

---

[5]https://projector.tensorflow.org/

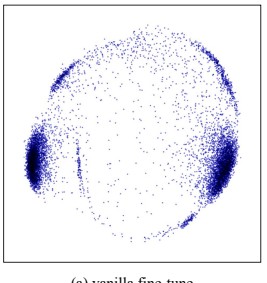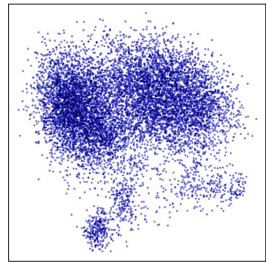

| (a) vanilla fine-tune | (b) our isotropy enhancement method |

Figure 6: PCA visualization of representations in vanilla fine-tuned XML-R and fine-tuned by our method.

| Model | Ar | En | Hi | Ru | Tr | Zh |
|---|---|---|---|---|---|---|
| LAS | | | | | | |
| XLM-R | 17.4 | 48.5 | 21.0 | 32.5 | 24.4 | 4.7 |
| +BN | 4.5 | 19.2 | 5.7 | 7.4 | 9.4 | 4.1 |
| +IsoBN | 4.8 | 18.1 | 6.4 | 7.7 | 10.7 | 3.8 |
| +ours | **23.6** | **54.6** | **21.7** | **34.6** | **26.9** | **5.8** |
| UAS | | | | | | |
| XLM-R | 32.4 | 55.7 | 37.5 | 44.3 | 41.2 | 17.7 |
| +BN | 15.8 | 32.1 | 19.3 | 19.0 | 24.3 | 14.3 |
| +IsoBN | 15.7 | 30.5 | 19.5 | 18.9 | 25.5 | 12.9 |
| +ours | **37.2** | **61.0** | **39.5** | **47.0** | **43.8** | **18.8** |

Table 3: Depprobe results on vanilla fine-tuned models and our proposed isotropy enhencement methods.

ple languages at the semantic level. As we saw in §2.2, many existing isotropic transformation methods do not preserve knowledge at the structural level, and we also conduct dependency syntax probe experiments on our proposed method with the same setting. Table 3 shows the LAS and UAS on XLM-R. Our proposed method can exceed the performance of the vanilla fine-tuning method on both metrics in all languages. In summary, the experimental results show that our proposed isotropy enhancement method enhances the isotropy of the representation distribution while maintaining the structural knowledge of the original representation. This may be because the distribution transformation by optimizing the Wasserstein distance can preserve some geometric characteristics of the original distribution (Panaretos and Zemel, 2019).

# 6 Related Work

**Anisotropy of Contextual Representation.** Nowadays, contextual representation (Peters et al., 2018; Devlin et al., 2019) often performs better than static representation in many NLP tasks. However, several studies (Gao et al., 2019; Ethayarajh, 2019; Cai et al., 2021) find contextual representations exhibit severe anisotropy in geometric properties. Their experiments show that

representations with high anisotropy negatively affect downstream tasks. Furthermore, Rajaee and Pilehvar (2022) find multilingual pre-trained language models have a higher degree of anisotropy than the corresponding monolingual ones.

Currently, a number of methods have been proposed to mitigate the degradation of contextual representations. Gao et al. (2019), Zhang et al. (2020) and Wang et al. (2020) tackle this problem by adding additional loss function constraints in the pre-training phase. In addition, there has been some work to enhance isotropy through post-processing methods. Li et al. (2020) utilize flow-based generative model to map contextual representations into isotropy standard normal distribution. Su et al. (2021) and Huang et al. (2021a) achieve the same effect by the whitening transformation. Rajaee and Pilehvar (2021a) propose a local cluster-based method. However, Rajaee and Pilehvar (2021b) find that though fine-tuning pre-trained language models can achieve a considerable performance boost, the representation space of fine-tuned models is still highly anisotropic. Unfortunately, many existing methods for adjusting the fine-tuned representation space for isotropy will hurt its performance. Our study finds that existing methods destroy the syntactic knowledge implied by original representations when applied to fine-tuning. In contrast, our proposed isotropy enhancement method can preserve as much important syntactic knowledge as possible in the fine-tuning stage.

**Zero-shot Cross-lingual Transfer.** Owing to the significant progress in multilingual pre-trained language models (mPLMs) (Devlin et al., 2019; Conneau and Lample, 2019; Conneau et al., 2020a; Xue et al., 2021; Chi et al., 2021, 2022; Scao et al., 2022b), zero-shot cross-lingual transfer achieves surprising performance in various NLP tasks (Hsu et al., 2019; Li et al., 2021; Sherborne and Lapata, 2022; Zheng et al., 2021).

However, to date, researchers have not been able to figure out what factors influence the ability of zero-shot cross-lingual transfer. Wu and Dredze (2019b) observe the sub-word overlap between the source and target languages has positive effects on the zero-shot performance. In contrast, some researchers (Pires et al., 2019; Karthikeyan et al., 2020) show no direct relationship between lexical overlap and cross-lingual transfer effects. de Vries et al. (2022) confirm the impact of some typological features, such as lexical-phonetic distances,

word order differences, and writing systems. Meanwhile, Chai et al. (2022) further find that word composition plays a more important role in cross-lingual transfer than other language properties. Additionally, some studies have focused on the representation level and proposed many methods to align representations and induce language-agnostic representations (Cao et al., 2020; Libovický et al., 2019; Zhao et al., 2021; Tanti et al., 2021). Huang et al. (2021b) considered the difference in representation between the source and target languages as noise in the contextual embedding and utilized robust training methods to tolerate noise.

## 7    Conclusion

In this paper, we propose a simple but effective method to improve the performance of zero-shot cross-lingual transfer. By introducing the isotropy enhanced fine-tuning and constrained code-switching, our proposed method can induce moderate isotropic representation and align multilingual representation. Experimental results on three zero-shot cross-lingual transfer tasks demonstrate the performance superiority of our method over existing methods. Extensive analytical experiments further confirm the effectiveness of our method for enhancing isotropic representations and reducing cross-lingual representation discrepancy.

## Limitations

Even though our work improves cross-lingual performance effectively, some limitations are still listed below:

- Our method is based on a widely accepted assumption: multilingual pre-trained language models can map the semantics of different languages to the same representation space, and representation alignment significantly affects cross-lingual generalization ability. However, a pre-trained model cannot cover all languages worldwide. For languages not seen in the pre-training stage, they may not be in the same representation space as the source language, and our method may have little effect.
- We conduct experiments on two strong masked language models. However, we have not successfully applied our method to the most promising generative pre-training models, such as BLOOM, and we will continue to explore in the future.
- Compared with the original fine-tuning, our method increases the training phase's time cost,

especially since calculating Wasserstein distance requires more computation. We will explore more efficient isotropy enhancement methods for cross-lingual transfer in the future.

## Acknowledgements

This work is supported by the National Science Foundation of China (NSFC No. 62206194), the Natural Science Foundation of Jiangsu Province, China (Grant No. BK20220488).

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

## A    The characteristics of datasets

Table 4 shows the detailed characteristics of datasets.

| Dataset | PAWS-X | XNLI | MARC |
|---|---|---|---|
| Task | Paraphrase | NLI | Sentiment |
| Class | 2 | 3 | 5 |
| \|Lang\| | 7 | 15 | 6 |
| Metric | Acc. | Acc. | Acc. |
| \|Train\| | 49,401 | 392,702 | 50,000 |
| \|Dev\| | 2,000 | 2,490 | 5,000 |
| \|Test\| | 2,000 | 5,010 | 5,000 |

Table 4: Characteristics of datasets

## B    Hyper-parameters

Table 5 shows the detail of hyper-parameters.

| | PAWS-X | | XNLI | | MARC | |
|---|---|---|---|---|---|---|
| | mBERT | XLM-R | mBERT | XLM-R | mBERT | XLM-R |
| lr | 2e-5 | 5e-6 | 1e-5 | 5e-6 | 2e-5 | 2e-6 |
| $\lambda_1$ | 0.5 | 0.5 | 0.5 | 1.0 | 0.5 | 0.5 |
| $\lambda_2$ | 1.0 | 1.0 | 1.0 | 1.0 | 1.0 | 1.0 |

Table 5: Details of hyper-parameters.

## C    Confidence interval

Table 6 shows the 95% confidence interval of the vanilla fine-tuning and our method.

| | PAWS-X | XNLI | MARC |
|---|---|---|---|
| mBERT | 83.51±0.07 | 66.34±0.48 | 45.80±2.00 |
| mBERT+ours | 85.29±0.29 | 67.20±0.15 | 47.90±0.78 |
| mBERT+DA+ours | 85.50±0.16 | 67.47±0.13 | 48.21±0.93 |
| XLM-R | 87.01±0.80 | 79.35±0.19 | 58.88±0.18 |
| XLM-R+ours | 89.03±0.13 | 80.90±0.56 | 59.71±0.13 |
| XLM-R+DA+ours | 89.48±0.28 | 81.40±0.08 | 59.89±0.12 |

Table 6: The 95% confidence interval of results.

## D    Results for each task and language

Table 7-9 show detailed results for each language in PAWS-X, XNLI and MARC.

| Models | en | de | es | fr | ja | ko | zh | **avg** |
|---|---|---|---|---|---|---|---|---|
| mBERT | 93.97 | 85.92 | 88.16 | 87.85 | 75.16 | 74.34 | 79.17 | 83.51 |
| mBERT+BN | 94.07 | 85.90 | 87.69 | 87.07 | 75.09 | 72.77 | 78.34 | 82.99 |
| mBERT+IsoBN | 93.98 | 85.76 | 88.12 | 87.64 | 74.97 | 73.77 | 79.14 | 83.34 |
| mBERT+CosReg | 93.96 | 85.75 | 88.60 | 87.70 | 74.28 | 73.30 | 78.67 | 83.18 |
| mBERT+NoisyTune | 93.69 | 86.59 | 88.22 | 88.02 | 76.42 | 74.23 | 79.85 | 83.86 |
| mBERT+DA | 93.47 | 88.3 | 88.27 | 88.33 | 77.90 | 77.04 | 80.71 | 84.86 |
| mBERT+ours | 94.16 | 87.53 | 89.70 | 89.35 | 78.06 | 77.30 | 80.91 | 85.29 |
| mBERT+DA+ours | 93.40 | 87.91 | 88.90 | 89.29 | 79.35 | 77.87 | 81.79 | 85.50 |
| XLM-R | 95.57 | 89.77 | 90.19 | 90.57 | 80.30 | 79.80 | 83.17 | 87.01 |
| XLM-R+BN | 95.46 | 90.38 | 90.82 | 91.02 | 80.63 | 80.25 | 83.59 | 87.45 |
| XLM-R+IsoBN | 95.42 | 90.64 | 90.86 | 90.75 | 81.51 | 80.66 | 83.36 | 87.60 |
| XLM-R+CosReg | 95.59 | 90.79 | 90.99 | 91.75 | 80.95 | 80.22 | 83.82 | 87.73 |
| XLM-R+NoisyTune | 95.39 | 90.58 | 90.90 | 91.17 | 80.80 | 80.61 | 83.26 | 87.53 |
| XLM-R+DA | 95.56 | 91.21 | 91.51 | 92.06 | 82.71 | 83.70 | 85.27 | 88.86 |
| XLM-R+ours | 95.64 | 91.37 | 91.98 | 92.61 | 83.19 | 83.15 | 85.25 | 89.03 |
| XLM-R+DA+ours | 96.09 | 92.00 | 92.13 | 92.59 | 83.66 | 84.26 | 85.60 | 89.48 |

Table 7: Detailed results in different languages on PAWS-X.

| Models | en | ar | bg | de | el | es | fr | hi |
|---|---|---|---|---|---|---|---|---|
| mBERT | 82.36 | 65.21 | 69.03 | 71.54 | 67.30 | 74.84 | 74.00 | 60.09 |
| mBERT+BN | 82.61 | 65.04 | 68.89 | 71.53 | 67.27 | 74.99 | 74.44 | 60.07 |
| mBERT+IsoBN | 82.38 | 64.98 | 69.25 | 71.46 | 67.37 | 74.55 | 74.06 | 60.56 |
| mBERT+CosReg | 82.68 | 64.89 | 68.36 | 71.55 | 67.17 | 74.48 | 73.99 | 59.91 |
| mBERT+NoisyTune | 82.73 | 65.03 | 68.89 | 71.72 | 67.05 | 74.57 | 74.07 | 60.74 |
| mBERT+DA | 81.63 | 64.73 | 68.48 | 71.22 | 66.39 | 74.77 | 73.71 | 60.08 |
| mBERT+ours | 82.73 | 65.47 | 68.74 | 72.34 | 67.67 | 75.67 | 74.72 | 61.62 |
| mBERT+DA+ours | 80.97 | 66.04 | 68.88 | 72.24 | 67.90 | 75.13 | 74.40 | 63.27 |
| XLM-R | 88.34 | 77.71 | 82.43 | 82.87 | 81.41 | 83.81 | 82.60 | 75.74 |
| XLM-R+BN | 88.60 | 77.99 | 82.42 | 82.90 | 81.53 | 83.92 | 82.61 | 75.82 |
| XLM-R+IsoBN | 88.52 | 78.30 | 82.63 | 82.90 | 81.79 | 84.08 | 82.75 | 75.95 |
| XLM-R+CosReg | 88.26 | 77.62 | 82.21 | 82.78 | 81.43 | 83.58 | 82.75 | 75.62 |
| XLM-R+NoisyTune | 88.61 | 78.11 | 82.97 | 82.96 | 81.80 | 83.96 | 82.75 | 76.20 |
| XLM-R+DA | 88.13 | 80.35 | 84.61 | 83.28 | 83.04 | 84.80 | 83.82 | 77.85 |
| XLM-R+ours | 89.13 | 80.33 | 84.06 | 83.55 | 82.84 | 85.22 | 83.79 | 77.90 |
| XLM-R+DA+ours | 88.42 | 81.07 | 84.83 | 83.44 | 83.78 | 85.10 | 84.49 | 79.05 |

| Models | ru | sw | th | tr | ur | vi | zh | **avg** |
|---|---|---|---|---|---|---|---|---|
| mBERT | 69.28 | 49.42 | 53.56 | 61.06 | 58.10 | 70.07 | 69.24 | 66.34 |
| mBERT+BN | 69.50 | 49.61 | 52.91 | 61.51 | 58.15 | 70.24 | 69.09 | 66.39 |
| mBERT+IsoBN | 69.46 | 49.71 | 54.25 | 61.82 | 58.64 | 70.82 | 69.69 | 66.60 |
| mBERT+CosReg | 68.92 | 50.19 | 53.36 | 61.41 | 57.83 | 70.27 | 69.19 | 66.28 |
| mBERT+NoisyTune | 69.45 | 49.04 | 54.02 | 61.00 | 58.14 | 70.57 | 69.58 | 66.44 |
| mBERT+DA | 69.60 | 48.66 | 53.89 | 60.70 | 59.78 | 70.58 | 70.58 | 66.32 |
| mBERT+ours | 69.15 | 51.35 | 55.03 | 63.57 | 58.68 | 71.80 | 70.55 | 67.27 |
| mBERT+DA+ours | 69.26 | 50.90 | 54.89 | 62.96 | 60.92 | 72.28 | 72.06 | 67.47 |
| XLM-R | 79.94 | 71.38 | 76.41 | 78.39 | 71.49 | 79.22 | 78.51 | 79.35 |
| XLM-R+BN | 79.90 | 71.59 | 76.71 | 78.26 | 71.60 | 79.35 | 78.70 | 79.46 |
| XLM-R+IsoBN | 80.20 | 71.87 | 76.92 | 78.67 | 71.90 | 79.65 | 78.77 | 79.66 |
| XLM-R+CosReg | 79.65 | 71.34 | 76.45 | 78.38 | 71.36 | 79.28 | 78.34 | 79.27 |
| XLM-R+NoisyTune | 80.18 | 72.17 | 77.20 | 78.89 | 71.93 | 79.34 | 78.88 | 79.73 |
| XLM-R+DA | 81.70 | 73.25 | 79.06 | 79.91 | 74.64 | 81.49 | 80.57 | 81.10 |
| XLM-R+ours | 81.10 | 72.92 | 78.24 | 79.98 | 73.70 | 81.08 | 79.72 | 80.90 |
| XLM-R+DA+ours | 81.88 | 72.40 | 79.42 | 79.45 | 75.02 | 81.77 | 81.11 | 81.40 |

Table 8: Detailed results in different languages on XNLI.

| Models | en | de | es | fr | ja | zh | avg |
|---|---|---|---|---|---|---|---|
| mBERT | 61.37 | 45.39 | 44.81 | 46.86 | 39.15 | 37.22 | 45.80 |
| mBERT+BN | 61.69 | 44.52 | 42.88 | 44.71 | 38.63 | 34.15 | 44.43 |
| mBERT+IsoBN | 61.65 | 44.85 | 44.38 | 46.27 | 38.11 | 37.02 | 45.38 |
| mBERT+CosReg | 62.75 | 47.51 | 45.91 | 46.88 | 39.79 | 37.48 | 46.72 |
| mBERT+NoisyTune | 61.44 | 46.60 | 45.23 | 47.06 | 39.57 | 38.15 | 46.34 |
| mBERT+DA | 61.69 | 48.40 | 46.70 | 47.22 | 39.26 | 38.61 | 46.98 |
| mBERT+ours | 63.25 | 48.99 | 47.94 | 48.18 | 39.68 | 39.36 | 47.90 |
| mBERT+DA+ours | 62.60 | 50.58 | 48.14 | 48.51 | 39.84 | 39.63 | 48.21 |
| XLM-R | 65.64 | 63.62 | 57.68 | 58.34 | 55.13 | 52.85 | 58.88 |
| XLM-R+BN | 65.90 | 64.35 | 57.82 | 58.54 | 56.03 | 53.10 | 59.29 |
| XLM-R+IsoBN | 65.78 | 63.98 | 57.58 | 58.74 | 55.88 | 53.30 | 59.21 |
| XLM-R+CosReg | 66.23 | 64.29 | 58.22 | 58.93 | 55.66 | 53.43 | 59.46 |
| XLM-R+NoisyTune | 65.41 | 63.90 | 57.72 | 58.28 | 56.06 | 52.57 | 58.99 |
| XLM-R+DA | 66.62 | 64.37 | 58.54 | 58.80 | 54.68 | 53.87 | 59.48 |
| XLM-R+ours | 66.53 | 64.54 | 58.90 | 59.22 | 56.28 | 52.81 | 59.71 |
| XLM-R+DA+ours | 66.54 | 65.15 | 58.84 | 58.97 | 56.77 | 53.08 | 59.89 |

Table 9: Detailed results in different languages on MARC.