# OpenReview forum: "Isotropic Representation Can Improve Zero-Shot Cross-Lingual Transfer on Multilingual Language Models"
_EMNLP/2023/Conference — EMNLP 2023 Findings_

### Official Review · Reviewer_AcYD · 2023-08-01

**Soundness:** 3

**Excitement:**

3: Ambivalent: It has merits (e.g., it reports state-of-the-art results, the idea is nice), but there are key weaknesses (e.g., it describes incremental work), and it can significantly benefit from another round of revision. However, I won't object to accepting it if my co-reviewers champion it.

**Paper Topic And Main Contributions:**

I was reviewing this paper for the ACL 2023 as well, so I will reuse relevant points here.

The paper proposes a multilingual language model fine-tuning methodology that aims to improve the anisotropy of the token representations. The method is based on a loss function that explicitly increases the isotropy and a code-switching methodology that improve the consistency of the representations. Authors fine-tune two popular multilingual LMs (mBERT, XLM-R) and observe how the crosslingual performance and representations differ for up to 7 different training methodologies, including the proposed one. They conclude that the proposed methodology is the best, yielding the best crosslingual performance, while managing to preserve syntactic properties of the model (as showcased by dependency parsing probing). Other techniques (visualization, CKA, etc) were also used to strengthen the claims.

**Reasons To Accept:**

- The proposed method seems to work and has beneficiary effects on both semantic and syntactic properties of the model.

- The proposed method is compared to other SOTA methods in Section 4.4 and is able to beat them all.

- The analysis section is quite strong and provide a lot of additional evidence for the proposed method.

**Reasons To Reject:**

- Overall, different sets of methods is used for different experiments, e.g., compare methods used in Section 2.2, 4.3 and 5.3. It is hard to make a conclusion about the other methods when we are not presented with a comprehensive overview of their behavior in different circumstances. The gap is especially noticeable for the isotropy measures in Section 5.3, because the only baseline used is CosReg and the results for I_cos are discarded.

- There is a =discrepancy in hyperparameter tuning budget for the baseline methods vs the proposed methods. If I understand it correctly, it has at least 4x more training runs as the baselines due to lambda_1 and lambda_2 tuning.

- Confidence intervals for the results are not reported. This is important because the results are calculated from N=5 runs and they are within 1-2% of each other.

**Reproducibility:**

4: Could mostly reproduce the results, but there may be some variation because of sample variance or minor variations in their interpretation of the protocol or method.

**Reviewer Confidence:**

3: Pretty sure, but there's a chance I missed something. Although I have a good feel for this area in general, I did not carefully check the paper's details, e.g., the math, experimental design, or novelty.

**Typos Grammar Style And Presentation Improvements:**

Figure 2 and 5 - This types of visualization with the points connected is better suited for time series.

In some cases, authors talk about "word embeddings". However, this term is usually used for non-contextual word representations (e.g. word2vec). In other cases they are talking about "token representations". Are these the same concepts?

The version of XLM-R is not reported (base vs large).

L321 - I don't think the models were released by HuggingFace

L140 W^T W is not an embedding matrix

L337 - What exactly are MUSE dictionaries. The paper references the following paper. Is that it? Conneau, G. Lample, M. Ranzato, L. Denoyer, and H. J´egou. Word translation without parallel data. arXiv:1710.04087, 2017.

L833 - Wrong accent in the name Libovický.

One appendix does not have a text at all, the other has a placeholder text only

---

> ### Author Rebuttal · Authors · 2023-08-29
>
> Thanks a lot for your valuable review comments. We hope the following response can address your concerns:
>
> - **Q1**: Different sets of methods is used for different experiments.
>
>   - **A1**: This suggestion is very helpful to improve the soundness of our paper, and due to rebuttal time constraints, for Fig. 3, we add the results of batch and IsoBN on mBERT, and the results on XLM-R, which we promise to give in the revised version.
> | $I_{cos}$ |de|en|fr|hi|ru|ar|sw|tr|vi|zh|
> | --- | --- | --- | --- | --- | --- | --- | --- | --- | --- | --- |
> |mbert | 0.27|0.25|0.27|0.31|0.28|0.30|0.32|0.30|0.29|0.28|
> |+cosreg|0.00|0.00|0.00|0.02|0.01|0.01|0.01|0.02|0.01|0.01|
> |+batch|0.27|0.26|0.28|0.30|0.28|0.30|0.29|0.29|0.29|0.28|
> |+isobn|0.24|0.21|0.24|0.27|0.25|0.26|0.25|0.25|0.25|0.25|
> |+ours|0.05|0.01|0.05|0.07|0.04|0.07|0.09|0.07|0.06|0.06|
> |log$I_{pc}$ |**de**|**en**|**fr**|**hi**|**ru**|**ar**|**sw**|**tr**|**vi**|**zh**|
> |mbert|-6.3|-5.0|-5.5|-4.5|-5.0|-4.7|-6.2|-6.2|-5.2|-6.2|
> |+cosreg|-5.5|-6.3|-6.1|-5.4|-6.2|-6.3|-6.4|-5.2|-5.7|-5.5|
> |+batch|-6.3|-5.1|-4.8|-2.6|-4.8|-2.5|-4.8|-4.8|-5.1|-4.7|
> |+isobn|-4.9|-4.8|-4.9|-5.0|-5.3|-5.1|-4.7|-4.9|-4.9|-4.9|
> |+ours|-1.5|-1.8|-2.4|-1.8|-1.3|-2.6|-2.7|-1.8|-1.9|-1.9|
>
> - **Q2**: The hyperparameter tuning budget is more than baseline methods.
>
>   - **A2**: Undeniably, our method introduces some hyperparameters compared with the vanilla method, but we didn't spend much energy on the hyperparameter search. $\lambda_1$ and $\lambda_2$ are only carried out in {0.5, 1}. Meanwhile, we find that the experimental results of our method are more robust to these two hyperparameters within a reasonable range of values. Therefore, we think that in practical applications, our method does not need to spend too much time on the fine search of hyperparameters.
> |$\lambda_1$ & $\lambda_2$|En|De|Es|Fr|Ja|Ko|Zh|Avg|
> | --- | --- | --- | --- | --- | --- | --- | --- | --- |
> |0.1 & 0.1|95.90|91.25|91.80|92.35|82.20|82.65|84.85|88.71|
> |0.5 & 0.5|95.76|91.13|91.75|92.54|82.30|83.06|85.31|88.84|
> |0.5 & 1.0|95.64|91.37|91.98|92.61|83.19|83.15|85.25|89.03|
> |1.0 & 0.5|95.76|91.39|91.88|92.51|82.63|82.85|85.18|88.88|
> |1.0 & 1.0|95.78|91.16|91.91|92.64|82.51|83.05|85.33|88.91|
> |5.0 & 5.0|95.65|90.44|91.08|91.68|81.09|80.86|84.54|87.90|
>
> - **Q3**: Confidence intervals for the results are not reported.
>
>   - **A3**: Thank you for highlighting the importance of reporting confidence intervals for our results. We apologize for not including this crucial information in our initial submission. To rectify this oversight, we report the 95% confidence interval of our method below and we will report confidence intervals for all experiments in Table 1 in our revised version.
> ||PAWSX|XNLI|MARC|
> | --- | --- | --- | --- |
> |mbert+ours|85.29$\pm$0.29|67.20$\pm$0.15|47.90$\pm$0.78|
> |mbert+da+ours|85.50$\pm$0.16|67.47$\pm$0.13|48.21$\pm$0.93|
> |xlmr+ours|89.03$\pm$0.13|80.90$\pm$0.56|59.71$\pm$0.13|
> |xlmr+da+ours|89.48$\pm$0.28|81.40$\pm$0.08|59.89$\pm$0.12|
>
> - Typos and presentation improvements:
>
> 1. Regarding the concept of word embedding you mentioned, we are sorry for the misunderstanding we caused. Like Timkey and Schijndel (2021), we consider that word embedding includes contextual and non-contextual word representation. In order to avoid misunderstanding, we will replace ''word embedding'' with the precise ''contextual word/token representation''.
>
> 2. The XLM-R we use is the xlm-roberta-large, and we will make this clear in the revised paper.
>
> 3. The MUSE dictionaries is released by Conneau et.al, 2017
>
> We surely will address other presentation issues and revise our paper accordingly. We are looking forward to seeing more inspiring feedback!
>
> References:
>
> [1] Timkey W, van Schijndel M. All Bark and No Bite: Rogue Dimensions in Transformer Language Models Obscure Representational Quality. EMNLP 2021: 4527-4546.
>
> [2] Conneau, G. Lample, M. Ranzato, L. Denoyer, and H. J´egou. Word translation without parallel data. arXiv:1710.04087, 2017.

---

### Official Review · Reviewer_7ZXD · 2023-08-06

**Soundness:** 4

**Excitement:**

4: Strong: This paper deepens the understanding of some phenomenon or lowers the barriers to an existing research direction.

**Paper Topic And Main Contributions:**

The paper explores isotropic representations for zero-shot cross lingual transfer. Anisotropic representations are defined as occupying a narrow cone in the vector space while isotropic ones are uniformly distributed. It is hypothesized that isotropic representations are helpful for multilingual models as they ensure better distribution of the word representations. The authors introduce a loss enforcing the isotropic property with a Wasserstein distance to a perfect isotropic distribution (normal distribution), this is combined with a "consistency" loss that enforces (sentence s, code switched s) pairs to be close by cosine distance (code switched s is synthesized with a bilingual dictionary). Results on 3 classification tasks (PAWS-X, XNLI and MARC) show improved performance over previous work.

**Reasons To Accept:**

* The paper is well written and easy to follow
* The method seems intuitive, code will be released (attached as supplementary) for easy reproducibility
* Ablations are provided (combination of consistency and isotropic losses)
* Additional analysis are provided, ie isotropy enhances syntactic properties of the model (table 3)


**Reasons To Reject:**

* The results for MARC are reported using 25% of the training data and might not carry to the full data (the authors state they did not have enough compute resources), though for other datasets the authors use the full data so the issue is minor

**Reproducibility:**

5: Could easily reproduce the results.

**Reviewer Confidence:**

4: Quite sure. I tried to check the important points carefully. It's unlikely, though conceivable, that I missed something that should affect my ratings.

---

> ### Author Rebuttal · Authors · 2023-08-29
>
> Thanks for your review comments and acknowledgement.
>
> Due to limited computational resources and the time constraints during the rebuttal phase, we are unable to complete the experiments on the entire MARC dataset, and if the final paper is accepted, we will report the relevant experiments in the appendix of the camera ready version.
>
> Thank you again for your recognition and suggestions for our work.

---

### Official Review · Reviewer_u81d · 2023-08-08

**Soundness:** 3

**Excitement:**

4: Strong: This paper deepens the understanding of some phenomenon or lowers the barriers to an existing research direction.

**Missing References:**

Exploring Anisotropy and Outliers in Multilingual Language Models for Cross-Lingual Semantic Sentence Similarity, Hammerl et al. 2023, https://aclanthology.org/2023.findings-acl.439.pdf

This paper is very recent but deals with similar topics and has relatively similar conclusions. I wanted to make sure that you were aware of it.

**Paper Topic And Main Contributions:**

This paper introduces a novel fine-tuning method to improve downstream performance of multilingual models. The approach is based on two auxiliary losses. The first loss pushes isotropy by penalizing the Wasserstein distance between the in-batch output distribution and a normal Gaussian. The second loss enforces a high cosine-similarity between original and code-switched versions of input texts. The authors run fine-tuning experiments with two steps in which the classification loss, the isotropy loss and the code-switched loss are weighted differently.
Extensive downstream results are presented for several datasets across various languages. The method seems to outperform related work by a substantial margin. The authors also run an ablation study and experiments to measure the syntactical performance of their models, and argue that the combination of their two losses allows isotropic distributions that don't discard syntactic information.

**Questions For The Authors:**

1 - Although it is part of state-of-the-art baselines in the context of zero-shot cross-lingual transfer, it seems to me a bit confusing that you compare to the DA baseline. Your method seems perfectly compatible with theirs, and motivated by different purposes. As you are neither claiming or seeking for the state-of-the-art results *for zero-shot cross-lingual transfer in general* in this study, why did you include a comparison with this orthogonal method?

2 - What are the final hyperparameters that performed the best in your search? In the paper, I could only find ranges, which in my opinion could harm reproducibility.

3 - Did the choice of $\lambda_1$ and $\lambda_2$ matter? Would it make sense to try more extreme (<<1 or >1) values?

4 - Several times along the paper, it is argued that isotropy regularization methods can degrade syntactical/downstream performance (e.g. L534-537). I may have missed it, but I could not find a reference for this in your paper. Are you referring to the Stable Anisotropic Regularization (https://arxiv.org/pdf/2305.19358.pdf) paper?

5 - I feel that the point made in L537-539 ("existing methods destroy the syntactic knowledge implied by original representations when applied to fine-tuning[, and ours didn't]") is not convincingly addressed in the paper. Did you run experiments for Table 3 on the ablation models? If using the isotropy loss only degraded the performance, do you think it would consolidate this claim?

**Reasons To Accept:**

The paper is well motivated and addresses a clear research problem. The method is novel and sensible. The experiments are thorough and the results are very promising. The authors successfully make the representation space more isotropic according to two metrics by using the Wasserstein distance, which may pave the way for further investigation of this technique in other works.
This paper is also relevant in the current discussion about whether isotropy is actually a desirable property of language models (https://arxiv.org/pdf/2305.19358.pdf), as it proves the point that CosReg can be outperformed both in isotropy enhancement and downstream performance.

**Reasons To Reject:**

A weakness of the paper is the lack of convincing explanation of the reason why these two losses work well together. Considering that isotropic regularization alone already outperforms most of the presented baselines, it is not clear why a second loss is really required to prove the main point of the paper. The authors argue that their study shows how "existing methods destroy the syntactic knowledge implied by original representations when applied to fine-tuning", although they present no empirical proof of this fact. For instance, it would have been very convincing if this comment was supported by an evaluation of the syntactic capabilities of all isotropy-enhancing methods in Table 3.

The DA baseline is not really relevant in my opinion as it is somehow orthogonal to the submitted article, and can be combined with the method as shown in the tables. It is also not clear if the two-phase approach is required, as it is not proven in the ablation study.

A weakness of the method itself, which is discussed in the limitations section, could be the overhead induced by the computation of the auxiliary losses. Depending on the batch size used and the embedding dimensions, the Wasserstein distance may be computationally costly, and it would be relevant to see the extent of this drawback measured in this paper.

**Reproducibility:**

4: Could mostly reproduce the results, but there may be some variation because of sample variance or minor variations in their interpretation of the protocol or method.

**Reviewer Confidence:**

4: Quite sure. I tried to check the important points carefully. It's unlikely, though conceivable, that I missed something that should affect my ratings.

**Typos Grammar Style And Presentation Improvements:**

### Typos
L113: "even uniformly" => "evenly" or "uniformly"?
L251: which control**s**
L558: researcher**s**

### Style
L77: "impedes" is a bit strong, as the anisotropic XLM-R still has cross-lingual capacity. Maybe "damages" or "hurts" would be better?

---

> ### Author Rebuttal · Authors · 2023-08-29
>
> Thanks for your valuable review comments and constructive suggestions.
> We hope the information attached below can mitigate your concern and the revision plan can make this paper more soundful.
>
> - **Q1**: The necessity of two-stage training is unclear.
>
>   - **A1**: We have tried adding both losses simultaneously, but it was unsatisfactory. We split training into two stages for two reasons. First, these losses may interfere with each other because isotropy enhancement widens the distance between representations so that representations are distributed more evenly, while constrained code-switching makes multilingual representations with the same semantics close. When optimizing, achieving a delicate balance is challenging. Also, simultaneous optimizations add more cost (equation 6 has more computational cost than equation 10). In order to dispel your concerns, we do ablation experiments on PAWSX, as shown in the table below. We can observe that although the one-stage training has a significant performance improvement compared to the baseline, it still lags behind the two-stage training.
> || En | De | Es | Fr | Ja | Ko | Zh | Avg |
> | --- | --- | --- | --- | --- | --- | --- | --- | --- |
> | xlm-r | 95.97 | 89.77|90.19|90.57|80.30|79.80|83.17|87.01|
> |xlm-r+ours|95.64|91.37|91.98|92.61|83.19|83.15|85.25|**89.03**|
> |one-stage|95.78|91.00|91.46|92.29|82.06|81.50|85.41|88.50|
>
> - **Q2**: The empirical proof of the syntactic is insufficient.
>
>   - **A2**: Your constructive suggestion is very helpful to improve the soundness of our paper. We have reported the syntactic performance of three isotropy-enhancing methods in Section 2.2 (Figure 2). We complement the performance of BN and IsoBN as follows:
> |Model| Ar | En | Hi | Ru | Tr | Zh |
> | --- | --- | --- | --- | --- | --- | --- |
> |LAS|
> | XLM-R | 17.4|48.5|21.0|32.5|24.4|4.7|
> |+BN|4.5|19.2|5.7|7.4|9.4|4.1|
> |+IsoBN|4.8|18.1|6.4|7.7|10.7|3.8|
> |+ours|**23.6**|**54.6**|**21.7**|**34.6**|**26.9**|**5.8**|
> |UAS|
> | XLM-R | 32.4|55.7|37.5|44.3|41.2|17.7|
> |+BN|15.8|32.1|19.3|19.0|24.3|14.3|
> |+IsoBN|15.7|30.5|19.5|18.9|25.5|12.9
> |+ours|**37.2**|**61.0**|**39.5**|**47.0**|**43.8**|**18.8**|
>
> - **Q3**: Why you compare to the DA baseline?
>
>   - **A3**: We hope that our approach has the practicality to further promote the cross-lingual transfer capability of mPLMs, and DA is one of the strongest baselines currently available, so we compare with it and find that our method can be compatible with it to achieve better cross-lingual transfer performance.
>
> - **Q4**: What are the final hyperparameters that performed the best in your search?
>   - **A4**: To ensure the reproducibility of our method, we report the final hyperparameters in the table below:
> | | PAWSX |  | XNLI |  | MARC |  |
> | --- | --- | --- | --- | --- | --- | --- |
> ||mBERT|XLM-R|mBERT|XLM-R|mBERT|XLM-R|
> |lr | 2e-5 | 5e-6|1e-5|5e-6|2e-5|2e-6|
> |$\lambda_1$|0.5|0.5|0.5|1.0|0.5|0.5|
> |$\lambda_2$|1.0|1.0|1.0|1.0|1.0|1.0|
>
> - **Q5**: Does the choice of $\lambda_1$ and$\lambda_2$ matter?
>   - **A5**: We report a more detailed study about the choice of$\lambda_1$ and$\lambda_2$. We can observe that our method is relatively robust to$\lambda_1$ and$\lambda_2$, but extreme (<<1 or >1) values will negatively affect performance.
> | $\lambda_1$ & $\lambda_2$| En | De | Es | Fr | Ja | Ko | Zh | Avg |
> | --- | --- | --- | --- | --- | --- | --- | --- | --- |
> | 0.1 & 0.1 | 95.90|91.25|91.80|92.35|82.20|82.65|84.85|88.71|
> |0.5 & 0.5|95.76|91.13|91.75|92.54|82.30|83.06|85.31|88.84|
> |0.5 & 1.0|95.64|91.37|91.98|92.61|83.19|83.15|85.25|**89.03**|
> |1.0 & 0.5|95.76|91.39|91.88|92.51|82.63|82.85|85.18|88.88|
> |1.0 & 1.0|95.78|91.16|91.91|92.64|82.51|83.05|85.33|88.91|
> |5.0 & 5.0|95.65|90.44|91.08|91.68|81.09|80.86|84.54|87.90|
>
> - **Q6**: What are the references on isotropy regularization methods can degrade syntactical/downstream performance?
>   - **A6**: We mention the references in Section 2.2 (Line 152-162), the corresponding references are Rajaee and Pilehvar, 2021b and Zhang, 2022. Regarding the missing reference you mentioned, we think it is highly relevant to our topic and we will cite it in the revised version.
>
> References:
>
> [1] Sara Rajaee and Mohammad Taher Pilehvar. How Does Fine-tuning Affect the Geometry of Embedding Space: A Case Study on Isotropy. findings of EMNLP, 2021
>
> [2] Zhang H, Liang H, Zhang Y, et al. Fine-tuning Pre-trained Language Models for Few-shot Intent Detection: Supervised Pre-training and Isotropization. NAACL, 2022: 532-542.

---

### Meta-Review · Area_Chair_XrdZ · 2023-09-19

**Recommendation:** 4

**Metareview:**

We thank the authors for their submission and engagement in the rebuttal.

The paper presents a two-step fine-tuning method for better zero-shot cross-lingual transfer.
The first step optimizes an isotropy-aware objective. The second step consists of code-switched data representation alignment based on cosine similarity.

Reasons to Accept:
- The introduced method improves cross-lingual transfer on three tasks and for two models.
- Analysis of the internal representation pre- and post-isotropy enhancement supports the method's effectiveness.

Reasons to Reject:
- The boost in performance for some languages is minor and would benefit confidence interval reporting.

---

### Decision · Program_Chairs · 2023-10-07

**Decision:**

Accept-Findings

**Comment:**

We thank the authors for their submission and engagement in the rebuttal.

The paper presents a two-step fine-tuning method for better zero-shot cross-lingual transfer.
The first step optimizes an isotropy-aware objective. The second step consists of code-switched data representation alignment based on cosine similarity.

Reasons to Accept:
- The introduced method improves cross-lingual transfer on three tasks and for two models.
- Analysis of the internal representation pre- and post-isotropy enhancement supports the method's effectiveness.

Reasons to Reject:
- The boost in performance for some languages is minor and would benefit confidence interval reporting.